# Developing a Predictive Grading Model for Children with Gliomas Based on Diffusion Kurtosis Imaging Metrics: Accuracy and Clinical Correlations with Patient Survival

**DOI:** 10.3390/cancers14194778

**Published:** 2022-09-29

**Authors:** Ioan Paul Voicu, Antonio Napolitano, Massimo Caulo, Francesco Dotta, Eleonora Piccirilli, Maria Vinci, Francesca Diomedi-Camassei, Lorenzo Lattavo, Alessia Carboni, Evelina Miele, Antonella Cacchione, Andrea Carai, Paolo Tomà, Angela Mastronuzzi, Giovanna Stefania Colafati

**Affiliations:** 1Department of Imaging, Bambino Gesù Children’s Hospital, IRCCS, 00165 Rome, Italy; 2Department of Imaging, “G. Mazzini” Hospital, 66100 Teramo, Italy; 3Medical Physics Unit, Risk Management Enterprise, Bambino Gesù Children’s Hospital, IRCCS, 00165 Rome, Italy; 4Department of Neuroscience, Imaging and Clinical Sciences, University G. d’Annunzio, 66100 Chieti, Italy; 5Department of Onco-Haematology, Cell and Gene Therapy, Bambino Gesù Children’s Hospital, IRCCS, 00165 Rome, Italy; 6Department of Laboratories, Pathology Unit, Bambino Gesù Children’s Hospital, IRCCS, 00165 Rome, Italy; 7Department of Radiology, Careggi University Hospital, 50134 Florence, Italy; 8Department of Neuroscience and Neurorehabilitation, Neurosurgery Unit, Bambino Gesù Children’s Hospital, IRCCS, 00165 Rome, Italy

**Keywords:** child, glioma, magnetic resonance imaging, diffusion tensor imaging, neoplasm grading

## Abstract

**Simple Summary:**

The most frequent brain tumors in children are solid tumors. A significant fraction of pediatric brain tumors is represented by gliomas, which are heterogeneous. Diffusion kurtosis imaging metrics (MK, AK, RK, FA, and ADC) have shown promising results for glioma grading in adult patients; however, it is unclear whether this technique is accurate for diagnosing high grade pediatric gliomas and if it is correlated with patient survival. In our study, we performed a retrospective whole-tumor analysis on 59 children affected by gliomas and tested (1) if DKI metrics are accurate for grading pediatric gliomas and (2) if DKI metrics are correlated with patient overall survival and progression-free survival.

**Abstract:**

Purpose: To develop a predictive grading model based on diffusion kurtosis imaging (DKI) metrics in children affected by gliomas, and to investigate the clinical impact of the predictive model by correlating with overall survival and progression-free survival. Materials and methods: 59 patients with a histological diagnosis of glioma were retrospectively studied (33 M, 26 F, median age 7.2 years). Patients were studied on a 3T scanner with a standardized MR protocol, including conventional and DKI sequences. Mean kurtosis (MK), axial kurtosis (AK), radial kurtosis (RK), fractional anisotropy (FA), and apparent diffusion coefficient (ADC) maps were obtained. Whole tumour volumes (VOIs) were segmented semi-automatically. Mean DKI values were calculated for each metric. The quantitative values from DKI-derived metrics were used to develop a predictive grading model to develop a probability prediction of a high-grade glioma (pHGG). Three models were tested: DTI-based, DKI-based, and combined (DTI and DKI). The grading accuracy of the resulting probabilities was tested with a receiver operating characteristics (ROC) analysis for each model. In order to account for dataset imbalances between pLGG and pHGG, we applied a random synthetic minority oversampling technique (SMOTE) analysis. Lastly, the most accurate model predictions were correlated with progression-free survival (PFS) and overall survival (OS) using the Kaplan–Meier method. Results: The cohort included 46 patients with pLGG and 13 patients with pHGG. The developed model predictions yielded an AUC of 0.859 (95%CI: 0.752–0.966) for the DTI model, of 0.939 (95%CI: 0.879–1) for the DKI model, and of 0.946 (95%CI: 0.890–1) for the combined model, including input from both DTI and DKI metrics, which resulted in the most accurate model. Sample estimation with the random SMOTE analysis yielded an AUC of 0.98 on the testing set. Model predictions from the combined model were significantly correlated with PFS (25.2 months for pHGG vs. 40.0 months for pLGG, *p* < 0.001) and OS (28.9 months for pHGG vs. 44.9 months for pLGG, *p* < 0.001). Conclusions: a DKI-based predictive model was highly accurate for pediatric glioma grading. The combined model, derived from both DTI and DKI metrics, proved that DKI-based model predictions of tumour grade were significantly correlated with progression-free survival and overall survival.

## 1. Introduction

Brain tumors are the most frequent solid tumors in children [1]. A significant fraction of pediatric brain tumors (up to 40%) is represented by gliomas [2]. Pediatric gliomas are heterogeneous, including both low-grade gliomas (pLGG) and high-grade gliomas (pHGG), and present peculiar histological and molecular characteristics based on the tumor anatomic location [3,4]. pLGGs are the most frequent pediatric gliomas and typically present as slow growing lesions [3]. They are different from IDH-mutant LGGs occurring in adults, because they rarely undergo malignant transformation and show excellent overall survival with the currently available treatments [3]. pHGGs are also different from their adult counterparts, both in terms of their diverse anatomical locations and underlying molecular drivers [5].

Magnetic resonance imaging (MRI) remains the gold standard for noninvasive glioma diagnosis. However, MRI glioma grading remains difficult in children, due to low numbers and significant histological and molecular heterogeneity. Diffusion kurtosis imaging (DKI) is a versatile diffusion technique [6] that measures the non-Gaussianity of water diffusion (via metrics such as mean kurtosis, radial kurtosis, and axial kurtosis), but also allows for obtaining typical DTI metrics based on a Gaussian model of water diffusion, such as fractional anisotropy or mean diffusivity (the equivalent of the apparent diffusion coefficient). DKI has shown promising results for glioma grading in adult patients [7,8,9]. A meta-analysis by Frank Delgado et al. [9] reported a pooled area under the curve of 0.94 for the discrimination of HGG from LGG in adults, with 0.85 (95% CI: 0.74, 0.92) sensitivity and 0.92 (95% CI: 0.81, 0.96) specificity. In all of these studies, DKI derived metrics were proven to be more accurate than DTI-derived metrics for grading purposes. However, as both DTI and DKI metrics can be calculated from the same diffusion sequence, it is possible that a combined approach, including information from both DTI and DKI metrics, could prove even more accurate and maximize the amount of useful information, which could be obtained from the DKI sequence.

The analysis model of DKI values used for adult gliomas implicitly assumes HGG to have higher DKI values than LGG, irrespective of the tumor location. Given that pediatric gliomas significantly differ from adult gliomas in terms of anatomic location and molecular drivers [2], it is unclear whether the same analysis model could translate to the landscape of pediatric gliomas and maintain the same accuracy reported in adult studies [9].

Differently from their adult counterparts, studies on pediatric glioma grading also face the issue of class imbalance between larger cohorts of pLGG and smaller cohorts of pHGG. This issue may also have limited the application of DKI models and accounts for the existing limited literature of DKI application in pediatric brain tumors [10], with a reported sensitivity of 80.8%, specificity of 85.7%, and AUC of 0.885 for a volume-based diffusion kurtosis.

In addition, so far, no studies explored the correlation between DKI metrics and patient survival.

The primary aim of our study was to investigate if DKI could be of added value in pediatric glioma grading, by developing a predictive model based on DKI metrics. A secondary aim was to assess its clinical impact by correlating the model predictions with patients’ overall survival and progression-free survival.

## 2. Materials and Methods

### 2.1. Clinical Data Collection

We identified 59 consecutive patients with histologically proven glioma [11] who underwent DKI on preoperative imaging between 1 February 2016 and 1 September 2020. For each patient, clinical data on patient overall survival (OS) and progression-free survival (PFS) were collected by two neuro-oncologists (E.M. and A.C.). Overall survival (OS) was defined as the length of time from the date of diagnosis to the date of death or last follow-up. Progression-free survival (PFS) was defined as the length of time from the date of treatment initiation until disease progression. Tumor progression was defined as radiological progression, associated with worsening of the pre-existing symptoms or the appearance of new symptoms.

### 2.2. Image Acquisition, Preprocessing, and Analysis

All patients underwent an identical MRI protocol on a 3 Tesla scanner (Siemens, Erlangen, Germany), including diffusion sequences. Patients unable to cooperate were imaged under sedation.

A prototype EPI-based diffusion-weighted sequence with blipped controlled aliasing for simultaneous multislice acquisition was exploited for data acquisition. Diffusion measurements were performed with dipolar diffusion sensitizing gradients applied in 30 directions with b values of 0 (10 averages), 1000, and 2000 s/mm^2^. The 10 averages were acquired only for 0 b-values. This approach was chosen to allow us a scan time, which would be feasible for clinical practice.

For each patient, standardized DKI model post-processing was implemented. The diffusion data analysis pipeline included motion correction [12] performed via Artefact correction in diffusion MRI (ACID) Matlab toolbox (http://www.diffusiontools.com/ (accessed on 22 September 2020)), whereas denoising, brain masking, and the diffusion kurtosis tensor estimation were performed via Mrtrix (http://www.mrtrix.org (accessed on 22 September 2020)). DKI-derived metrics estimation was computed as previously described [6,13]. All processing and metric estimations were then embedded in a batch Matlab function (MathWorks, Natick, MA, USA, version 9.2 R2017a). The pipeline output included five metrics: MK, AK, RK, FA, and MD/ADC; for the purpose of this study, the terms MD and ADC were used interchangeably, as MD is the mathematical equivalent of ADC, but is applied in more directions. Conventional sequences were then co-registered with DKI maps using an affine transformation in 3DSlicer.

Whole-tumor volumes were segmented semi-automatically on co-registered images with ITK-SNAP [14] by a neuro-radiologist (IPV, 7 years of experience) using the tissue classification and the active contour features (Figure 1). Contrast-enhancing and non-enhancing solid tumor regions were segmented separately. Necrotic and hemorrhagic regions, calcifications, and peri-tumoral edema were not included in the segmentation. Given the dependence of kurtosis parameters on the magnetic susceptibility [15], in order to adjust for possible age-related differences in T2* [16], DKI-derived metrics values were normalized to normal appearing white matter of the cerebral hemisphere unaffected by the tumor. The normal appearing white matter was segmented semi-automatically using the tissue classification tool available in ITK-SNAP and by limiting the iterations to 100. Regions of periventricular edema in patients with tumors determining obstructive hydrocephalus were carefully avoided. Two experienced neuro-radiologists (G.S.C. and M.C., both with 20 years of experience) reviewed the segmentations in consensus. Neuroradiologists were blinded to the histological diagnosis and clinical data. Pathologists were blinded to the DKI analyses and clinical data. Clinicians were blinded to the DKI analyses and histological diagnosis.

### 2.3. Statistical Analysis

An a priori power analysis was computed to estimate the required sample size for a statistical power of 80% with an alpha of 0.05 and a ratio of 2:1 between pLGG and pHGG, assuming an effect size of 0.8.

DKI-derived diffusion metrics underwent three levels of analysis.

First, differences between the means of pLGG and pHGG were tested with independent sample t tests, applying the Benjamini–Hochberg correction for multiple comparisons.

Second, metrics that presented significant differences underwent a receiver operating characteristic (ROC) analysis to assess the separate grading accuracy of each metric for grading purposes.

Third, all DKI-derived diffusion metrics were used to create predictive grading models based on penalized logistic regression with Elasticnet penalization (package glmnet, R), a penalty method based on a combination of LASSO and Ridge regression penalties useful to avoid overfitting of the model [17].

Finally, three predictive models were built: (i) a DTI-based model (based on FA and ADC); (ii) a DKI-based model (based on MK, RK and AK); and (iii) a combined prediction model (based on all of the available diffusion metrics obtained from the DKI sequence). The regression coefficients obtained from each model were used to create probabilities for each patient of being affected by a pLGG or a pHGG. The resulting probabilities were tested with ROC analysis to assess the overall predictive accuracy of each model. After assessing the respective accuracies, the most accurate model predictions were used to assess the optimal probability cutoff for grade prediction.

In order to exclude that our results may have been influenced by the imbalanced dataset, we further tested our dataset with the Random Synthetic Minority Oversampling Technique (SMOTE) analysis (package SMOTE from the library imblearn in Python). The SMOTE preprocessing algorithm is considered “de facto” the standard in the framework of learning from imbalanced data [18]: it carries out an oversampling approach to rebalance the original training set by introducing new synthetic examples. The procedure is focused on the “feature space” rather than on the “data space”, in other words, the algorithm is based on the values of the features and their relationship, instead of considering the data points as a whole [17].

The random forest classifier was constructed through testing various hyperparameters, and selecting the best ones in terms of their predictive performance on the training set. The hyperparameters were manually set as follows:-“number of estimators” = 100, which describes the number of trees in the model.-“maximum depth” = 3, which describes the maximum depth of each tree in the forest.

A random seed was selected to guarantee the reproducibility of the results among the different computers. Then, a random seed was selected, which ensured a similar result for the model if it was used on different computers.

Using SMOTE, 33 synthetic pHGG cases were created. Thus, a balanced dataset of 46 pLGG and 46 pHGG (13 real and 33 synthetic samples randomly created) was obtained, totaling a number of 92 cases. The final dataset was split into a 0.5 training set and testing set. Testing set accuracy was assessed by computing the confusion matrix and calculating the resulting AUC.

Finally, the clinical relevance of the predictive combined model was assessed with survival analysis. Model predictions were stratified as low-grade or high-grade, respectively, based on the previously calculated threshold. Survival metrics included the overall survival and progression-free survival.

Probability of OS and PFS were calculated according to the Kaplan–Meier method [19] and expressed as median OS and median PFS, respectively. All of the results were expressed as probability or cumulative incidence (%) and 95% confidence interval (95% CI). Differences in OS and PFS were estimated with the log-rank test (Mantel–Cox).

Statistical analyses were performed with SPSS (version 20), R Studio (version 1.1.463), and Python (version 3.9). Power analyses were performed with G*Power (version 3.1.9.3). *p* values < 0.05 were considered statistically significant.

Survival analysis used 1 April 2021 as the reference date.

## 3. Results

### 3.1. Patients Features

The a priori power calculation yielded a requested sample of 58 patients. All 59 patients (33 M, 26 F) received a histologic diagnosis of glioma. The cohort included 46 patients with pLGG and 13 patients with pHGG (Table 1). The median age at diagnosis was 7.2 years (IQR 5.0–11.5 years). Median PFS was 21 months (IQR: 10–36 months). Median OS was 23 months (IQR: 15–37 months). The DKI sequence acquisition parameters are listed in Table 2.

### 3.2. DKI Differences between pLGG and pHGG

DKI metrics presented significant differences between pLGG and pHGG for all metrics, with pLGG presenting significantly different values compared with pHGG for each metric (RK: 0.34 ± 0.1 vs. 0.65 ± 0.34, MK: 0.44 ± 0.9 vs. 0.72 ± 0.18, AK: 0.58 ± 0.9 vs. 0.85 ± 0.21, FA: 0.44 ± 0.16 vs. 0.57 ± 0.12, ADC: 2.07 ± 0.51 vs. 1.36 ± 0.54, *p* < 0.01 for all metrics). The ROC analysis yielded AUC values greater than 0.8 for each metric, with the exception of FA, with the most accurate metric being MK (AUC 0.9231 (95%CI: 0.840–1). The AUC values and CI for each metric are reported in detail in Figure 2.

### 3.3. Predictive Model Development

The model coefficients for each analysis model (DTI-only-based, DKI-only-based, and DTI and DKI, combined) are summarized in Table 3. The model coefficients were used to calculate the predicted probability of having a pHGG for each patient.

### 3.4. Model Accuracy Estimation

The predictive models presented different accuracies, with the DTI-only-based model yielding the lowest accuracy, the DKI-only-based model yielding an intermediate accuracy, and the combined DKI and DTI model proving to be the most accurate. Specifically, ROC tests of the estimated probabilities of the DTI, DKI, and combined predictive model yielded AUC values of 0.859 (95%CI: 0.752–0.966), 0.939 (95%CI: 0.879–1), and 0.946 (95%CI: 0.889–1), respectively. The ROC curves for each model are represented in Figure 3. Thus, the most accurate model proved to be the combined model. The combined model prediction threshold of 17.3% probability of having an HGG yielded 92.3% sensitivity and 82.6% specificity.

### 3.5. SMOTE Analysis

By means of the SMOTE analysis, a balanced dataset was obtained, comprising 46 pLGG and 46 pHGG (13 real and 33 synthetic samples). This balanced dataset of 92 cases was evenly divided into a training and testing cohort. A random-forest-based model classifier was trained, and then applied on the testing set of 46 patients, and it predicted all but one of the SMOTE-generated samples (Figure 4). The sample estimation with the random SMOTE technique yielded an AUC of 0.98.

### 3.6. Clinical Correlations of the Predictive Model

The whole-cohort survival analysis showed a significant correlation between the combined predictive model and both OS and PFS (*p* < 0.001 for both OS and PFS for all metrics, Figure 5 and Figure 6). Patients with a model prediction of pHGG presented both a significantly lower PFS and OS compared with patients with a model prediction of pLGG.

## 4. Discussion

In light of new molecular advances in brain tumors—especially in the pediatric population—the accurate assessment of the WHO grade is of the utmost importance and still relies on invasive tissue sampling. However, biopsies and incomplete resections may be hampered by a non-representative sample due to intra-tumoral heterogeneity.

Therefore, developing a pre-operative model predictive of glioma grading would be invaluable for the management and the proper risk-stratification of pediatric patients.

This is one of the first studies to explore the potentials of diffusion kurtosis imaging in pediatric brain tumors, and the first to investigate DKI for pediatric glioma grading and to correlate it with patient survival.

Our results support the initial hypothesis that DKI is of added value in pediatric glioma grading, having obtained good to excellent grading on accuracy-based information from DTI and DKI metrics, respectively, in a relatively large cohort. Such results are further supported by the SMOTE analysis, thus excluding that our results may have been influenced by the imbalanced dataset.

Moreover, our preliminary data showed that DKI metrics had a significant correlation with overall survival and progression-free survival (Figure 5 and Figure 6), similarly to tumor histology (Appendix A). In particular, patients with predicted pLGG presented significantly higher OS and PFS than patients with predicted pHGG. The results (*p* < 0.001 for both PFS and OS for all metrics) support the hypothesis that DKI is not only useful for glioma grading, but could also have a strong clinical impact on patient management, should these findings be confirmed on larger, possibly multicohort and prospective studies.

Studies on adult gliomas have previously reported DKI metrics to have superior accuracy compared with DTI and DWI metrics for grading purposes [5,7,8,20,21,22,23]: reflecting diffusional non-Gaussianity, it might provide better information on tissue structure that produces compartments [24]. In adults, HGG presented significantly higher values of DKI-derived metrics, such as axial kurtosis (AK), radial kurtosis (RK), and mean kurtosis (MK). This pattern has been explained as being related to a higher tumor heterogeneity in HGG compared with LGG [6].

Based on our data, these assumptions also hold true for pediatric gliomas with hemispheric and supratentorial midline locations (Figure 7 and Figure 8).

The KFA metric was not exploited in our model because of its poor performance [25], consistent with previous findings in adults [5,7,8]

She and colleagues [10] first retrospectively correlated DWI, IVIM-derived, and DKI metrics with histopathologic features and tested the accuracy of these metrics for grading in pediatric patients affected by brain tumors by means of both a ROI- and VOI-based approach. Although DKI was found to be a good estimator of tumor cellularity, they obtained similar grading performances for both DWI and DKI metrics.

In our study, we obtained a higher grading accuracy both for DKI metrics alone (AUC of 0.9231 for MK, 95%CI: 0.840–1) and for a combined prediction model including the information from all of the diffusion metrics obtained from the DKI sequence (AUC of 0.946 for the combined model, 95%CI: 0.889–1). In addition, our findings show DKI metrics to have superior accuracy compared with the DTI and DWI metrics, in line with previous studies on adults [5,7,8,9]. Aside from the differences in methodology and analysis, some of the differences from the two studies may have been due to a more homogeneous cohort in our case, where we specifically focused only on patients with proven glioma histology. Further studies should be performed to assess the role and clinical impact of DKI in pediatric brain tumors and tumor subtypes.

In our study, we analyzed the whole tumoral volume, including both tumoral contrast-enhancing and non-enhancing sub-regions. Contrast-enhancing and non-enhancing tumor regions had to be segmented separately, in order to maximize the accuracy tissue classification and the active contour features on ITK. We chose to analyze the whole solid tumoral volume rather than the solid tumoral contrast-enhancing and non-enhancing sub-regions for two reasons. First, pediatric gliomas are very heterogeneous and our cohort made no exception: some showed no contrast enhancement, others were exclusively contrast enhancing and others showed both contrast enhancing and non-enhancing components. As such, considering only tumoral sub-regions could have limited the possible comparisons between the cases.

Second, we believe model assumptions based on whole-tumor data, obtained from thousands of voxels, help extract the maximum amount of information available from the images and increase the robustness of model assumptions based on these data: indeed, they are able to capture the heterogeneity of the whole tumoral environment with good correspondence to the histologic samples, thus maximizing their potential clinical impact.

Infratentorial midline gliomas were deliberately excluded from our analysis. As the molecular and genomic landscape of pediatric gliomas were redefined [2,3,4,20,21,22], key differences with adult gliomas and specific location-related differences were unveiled. High-grade gliomas in this location, formerly known as DIPG, have been reclassified as diffuse midline gliomas, H3K27M mutated [11] and have a dismal prognosis. Research is still in progress towards a better understanding of this subgroup of gliomas [20,22,25] and their underlying molecular drivers. These tumors were reported to exhibit unusual radiologic features. Therefore, their DKI metrics could also differ from the DKI metrics of high-grade gliomas in other locations. Further research on wider cohorts is required in order to develop a robust, holistic, radiologic pediatric glioma-grading model, which fully accounts for location-related differences.

This retrospective study has some limitations. Despite the cohort being quite large for pediatric gliomas, the relatively low number of patients requires caution in the interpretation of these results: validation on wider cohorts, possibly in prospective studies, should be performed to validate the findings. Our analysis assumes that pHGG with hemispheric cerebellar locations had significantly higher DKI values than pLGG, but as neither the analysis nor the validation cohort contained pHGG with that location, further testing may disconfirm that assumption. This limitation was partially overcome by the very high accuracy of predicting a low tumor grade in this subgroup. Furthermore, incorporating DKI in multi-parametric data-driven analyses including perfusion metrics may yield a higher accuracy than using a single sequence [23]. In addition, performing a histogram analysis instead of comparing the means of DKI metrics could have been found in other analysis models, possibly including machine-learning approaches [24], which were already described in adult patients with gliomas, and could be even more accurate descriptors of pediatric glioma heterogeneity. Finally, compared with conventional DWI sequences, DKI sequences had a significantly longer duration, often requiring imaging under general anesthesia. Although our results suggest DKI could be helpful in achieving a broader understanding of these tumors and finding significant correlations with clinical outcomes, further studies are needed to definitely assess the clinical benefit of DKI sequences, which require pediatric patients to be exposed to a longer scan time.

## 5. Conclusions

By performing whole-tumor analysis, DKI-derived diffusion metrics proved helpful in creating an accurate grading model in pediatric gliomas (Appendix A). Such a model was also correlated significantly with patient overall survival and progression-free survival. This information could be used in a preoperative setting to better predict glioma grade in children, potentially helping to adjust subsequent clinical and surgical decision-making.

## Figures and Tables

**Figure 1 cancers-14-04778-f001:**
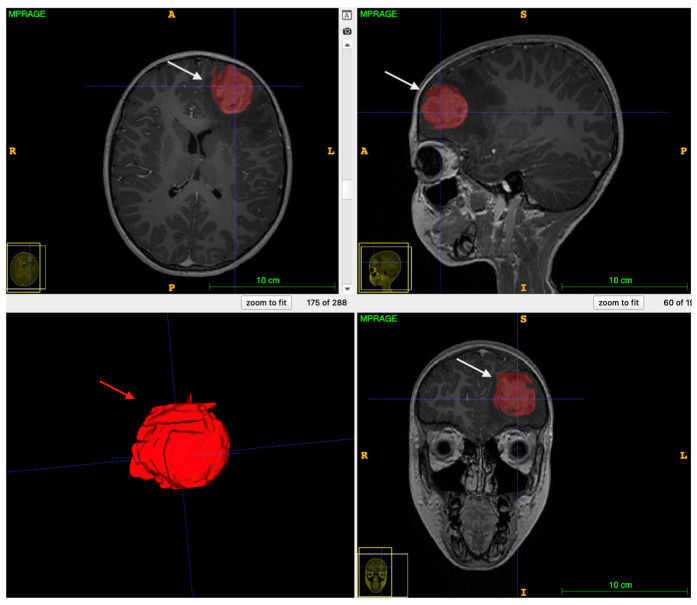
Semi-automatic segmentation and 3D tumor rendering with ITK-SNAP in a 9-year-old boy affected by a left frontal pHGG (grade 4 WHO). The lesion is characterized by a vividly enhanced solid tumor component on post-contrast multiplanar T1-weighted images (white arrows) surrounded by peritumoral edema. The solid tumor component (labeled in red) was segmented semi-automatically with the active contour feature. Semi-automatic segmentation allowed reconstruction of the whole tumor volume (red arrow).

**Figure 2 cancers-14-04778-f002:**
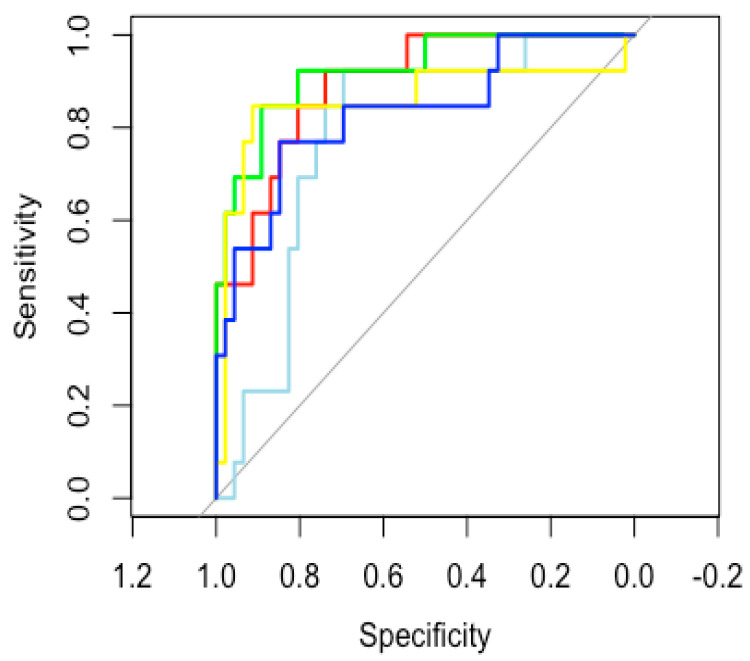
Glioma grading AUC values for each DKI metric. ROC analysis (pHGG = 1) yielded the following AUC values: MK (green curve): 0.9231 (95%CI: 0.840–1), AK (red curve): 0.8946 (95%CI: 0.806–0.982), RK (yellow curve): 0.8595 (95%CI: 0.706–1), FA (light blue curve): 0.7843 (95%CI: 0.656–0.912), ADC (blue curve): 0.8328 (95% CI: 0.698–0.967). DKI: diffusion kurtosis imaging; MK: mean kurtosis; AK: axial kurtosis. RK: radial kurtosis.

**Figure 3 cancers-14-04778-f003:**
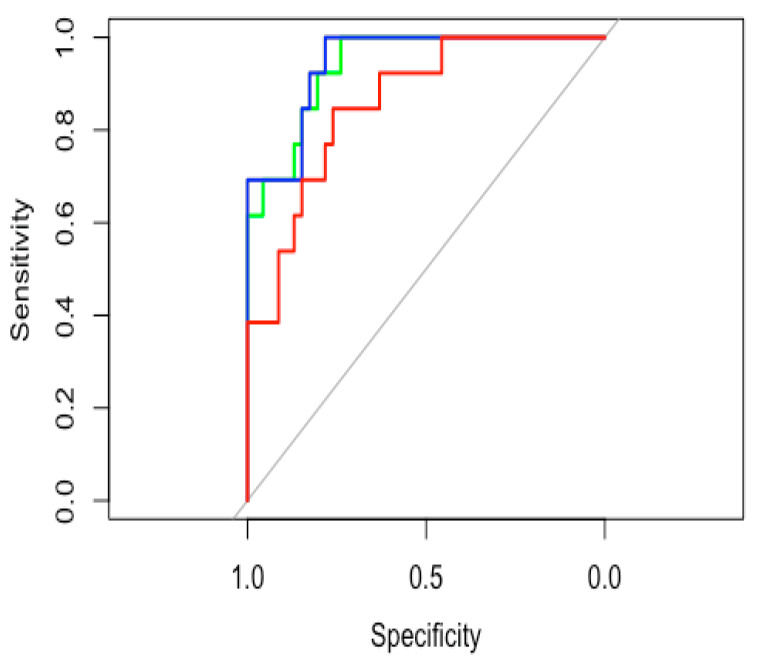
Glioma grading AUC values for the model predictions. Diffusion metrics from the DKI values were used to create model predictions of having a pLGG or a pHGG. ROC tests of the estimated probabilities of the DTI (red curve), DKI (green curve), and combined (DKI + DTI, blue curve) predictive model yielded AUC values of 0.859 (95%CI: 0.752–0.966), 0.939 (95%CI: 0.879–1), and 0.946 (95%CI: 0.889–1), respectively. pHGG: pediatric high-grade gliomas; pLGG: pediatric low-grade gliomas; DKI: diffusion kurtosis imaging.

**Figure 4 cancers-14-04778-f004:**
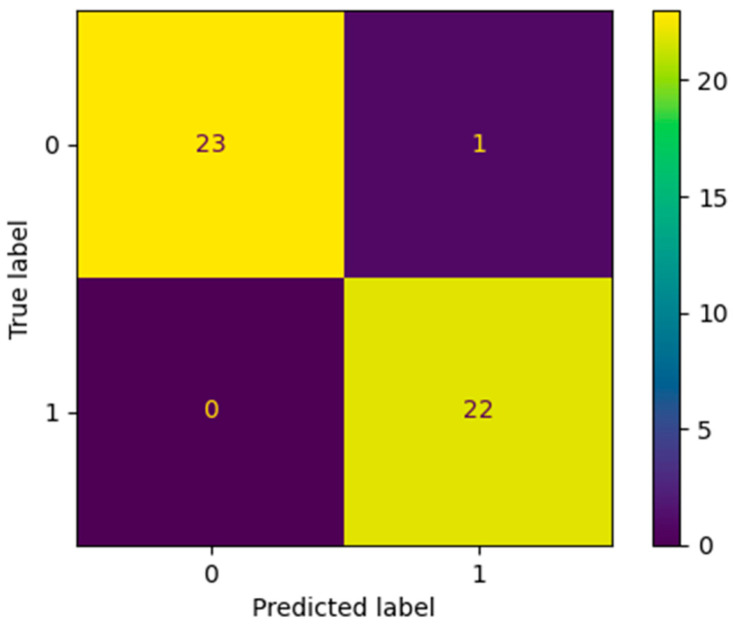
Confusion matrix of the random SMOTE analysis on the testing set of 46 patients (label 1 = pHGG, label 0 = pLGG). A model based on all DKI metrics correctly predicted all but one of the generated samples. DKI: diffusion kurtosis imaging.

**Figure 5 cancers-14-04778-f005:**
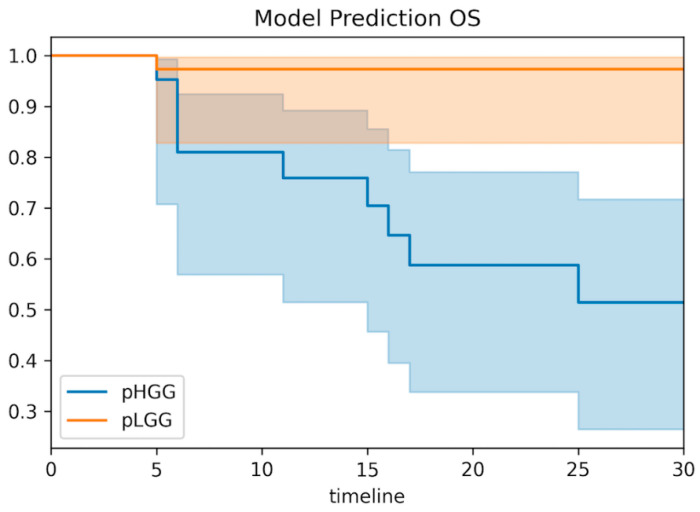
Model correlations with patient overall survival. Based on the combined predictive model, patients were classified as pHGG or pLGG based on the probability threshold of 17.3%. Kaplan–Meier estimates of model predictions presented significant correlations with patient overall survival (OS) (*p* < 0.001).

**Figure 6 cancers-14-04778-f006:**
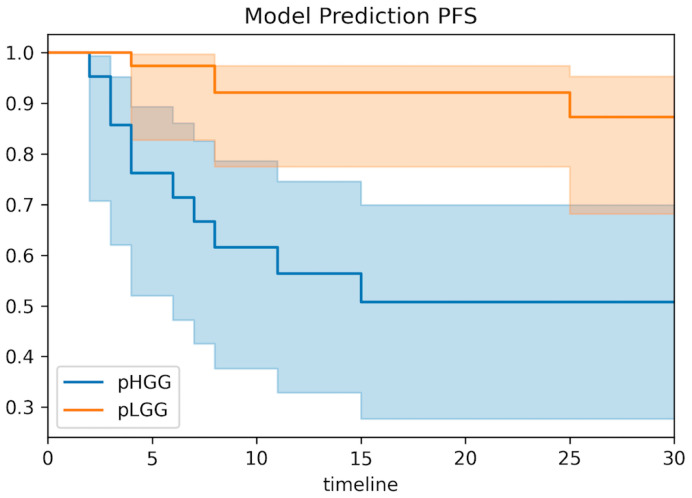
Model correlations with patient progression-free survival. Based on the combined predictive model, patients were classified as pHGG or pLGG based on the probability threshold of 17.3%. Kaplan–Meier estimates of model predictions presented significant correlations with patient progression-free survival (PFS) (*p* < 0.001).

**Figure 7 cancers-14-04778-f007:**
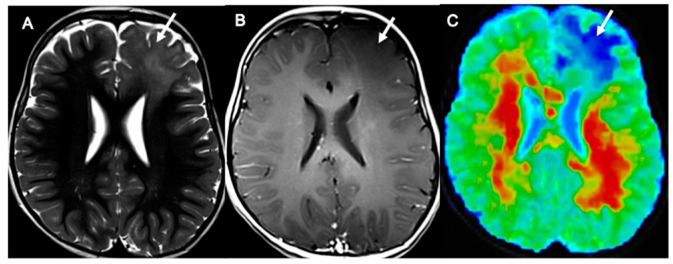
DKI values in pLGG. Axial TSE T2 sequences (**A**), T1 3D SPACE post-contrast sequences (**B**), and mean kurtosis color maps overlaid on T2 sequences (**C**) in a 4-year-old boy affected by a left frontal low-grade astrocytoma (grade 2 WHO). The pLGG ((**A**,**B**), arrows) presents low MK values ((**C**), arrows; MK of the tumor volume in this patient: 0.42). pLGG = pediatric low-grade glioma; MK = mean kurtosis.

**Figure 8 cancers-14-04778-f008:**
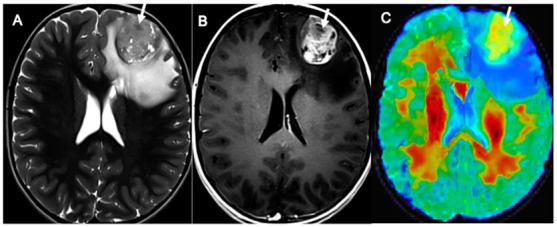
DKI values in pHGG. Axial TSE T2 sequences (**A**), T1 3D SPACE post-contrast sequences (**B**), and mean kurtosis color maps overlaid on T2 sequences (**C**) in a 9-year-old boy affected by a left frontal pHGG (grade 4 WHO, same patient as in Figure 1). The pHGG ((**A**,**B**), arrows) presents high MK values ((**C**), arrow; MK of the tumor volume in this patient: 0.70). pHGG: pediatric high-grade glioma; MK: mean kurtosis.

**Table 1 cancers-14-04778-t001:** Patient demographic, histologic, and molecular characteristics.

Patient Characteristic	Number of Patients
Demographic characteristics	59/59 (100%)
Median age	7.2 (5.0–11.5)
Sex	
Male	32/59 (55.1%)
Female	26/59 (44.8%)
Histologic characteristics	
Low-grade gliomas (WHO 1–2)	46/59 (79.3%)
High-grade gliomas (WHO 3–4)	13/59 (20.7%)
Anaplastic astrocytoma (WHO 3)	2/59 (3.4%)
Anaplastic ganglioglioma (WHO 3)	1/59 (1.7%)
Anaplastic Pleomorphic Xanthoastrocytoma (WHO 3)	1/59 (1.7%)
Astroblastoma (WHO 1)	1/59 (1.7%)
Diffuse midline glioma, H3K27Maltered (WHO 4)	2/59 (3.4%)
Dysembryoplastic neuroepithelial tumor DNET (WHO 1)	1/59 (1.7%)
Ganglioglioma (WHO 1)	14/59 (23.7%)
Glioblastoma, IDH wild type (WHO 4)	5/59 (8.5%)
Low-grade Astrocytoma (WHO 2)	4/59 (6.8%)
Pleomorphic Xanthoastrocytoma (WHO 2)	1/59 (1.7%)
Pilocytic Astrocytoma (WHO 1)	14/59 (23.7%)
PLNTY (Polymorphous low-grade neuroepithelial tumor of the young, WHO 1)	2/59 (3.4%)

**Table 2 cancers-14-04778-t002:** SMS diffusion sequence acquisition parameters.

Parameters	Values
Number of directions	30
B values (s/mm^2^)	0, 1000, 2000
Repetition Time TR (ms)	7400
Echo Time TE (ms)	113
Acquisition matrix	128 × 128
Resolution (mm^3^)	2 × 2 × 2
Average slice number	36
GRAPPA	2
Reference lines	38
Acquisition time (min:s)	12.00
Bandwidth (hz/px)	1562
Acceleration Factor	2

**Table 3 cancers-14-04778-t003:** The model coefficients for each analysis model (DTI-only based, DKI-only based, and combined DTI and DKI) are summarized in Table 3. The model coefficients were used to calculate the predicted probability of having a pHGG for each patient. pHGG: pediatric high-grade gliomas; DKI: diffusion kurtosis imaging.

Model Coefficients for DKI Predictive Grading.			
Feature Name	DTI	DKI	DTI + DKI
**Intercept**	1.574	−9.699	−11.412
**Fractional anisotropy (FA)**	2.743	-	3.927
**Apparent diffusion coefficient (ADC)**	−2.484	-	−0.275
**Radial Kurtosis (RK)**	-	−5.952	−5.474
**Mean Kurtosis (MK)**	-	24.046	19.904
**Axial Kurtosis (AK)**	-	−3.164	-

## Data Availability

The data presented in this study are available upon request from the corresponding author. The data are not publicly available due to ethical concerns.

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
