# Peer review of "Developing a Predictive Grading Model for Children with Gliomas Based on Diffusion Kurtosis Imaging Metrics: Accuracy and Clinical Correlations with Patient Survival"

_cancers, 2022, doi:10.3390/cancers14194778_

Round 1
Reviewer 1 Report
Glioma are currently among the most common brain oncological lesions affecting paediatric patients. Tumour grading proved to have an overall strong correlation with clinical outcomes and predictive image-based models were extensively investigated in adult patients. The authors aimed at the development of an MR DKI-based grading model in paediatric glioma patients and the assessment of its correlation with survival metrics. The results are promising and consistent with literature data on adult patients, although further studies should be carried out on larger samples covering different tumour sites to confirm such results. Biases were pointed out in the discussion section.
I believe that this work is of high clinical interest, considering the potential impact on paediatric patients. I’m in favour of publication with minor revisions.
The manuscript is well written, with an overall good readability. Here follows few minor issues that need to be clarified or adjusted:
- Line 21: DKI acronyms should be added in the simple summary.
- Considering the central importance of the statement I suggest to include a reference for the sentence line 62-63 (Introduction section).
- Line 114: it is not clear if the 10 averages were acquired only for 0 b-value, or also for all the others (1000 and 2000) as they are expected to be more noisy.
- Line 121: provide the Matlab version.
- Please clarify the segmentation method and clinical decisions to account or not account for contrast-enhancing and non-contrast enhancing tumour regions. In particular, the whole-tumour volumes were segmented (line 126), ….. , contrast-enhancing and non-contrast enhancing tumour regions were segmented separately. How did you consider these two different segmentations? If you used them please clarify and provide results for them, otherwise I suggest a removal.
- The clinical approach to segmentation did not seem fully consistent between method section (lines 128-130) and discussion section (lines 304-317). Please clarify.
- I suggest to include in the method section (lines 171-175), few characteristics of the predictive model (random forest) that was developed on the training set and applied in the testing set to derive the accuracy of such model. Also the input parameters should be mentioned (the one from the most significant model, the most significant metrics, all the DKI-derived metrics?).
- In the method section (line 176-179) and also in the abstract (line 37), you stated that the combined model was tested for correlations with survival metrics. Did you consider the most accurate model (as reported in 160-161) or did you consider a priori the combined model for such analysis? I suggest to accordingly adjust this paragraphs.
- Line 196: Table1, in bracket range or percentage.
- Lines 199-202: considering ADC one of the DKI-metrics (although a DTI-metric) it is not consistent with previous sentence as higher in pLGG rather than in pHGG.
- Figure 1, Figure 2, and Figure 4 should be consistent: True Positive rate vs False Positive Rate or Sensitivity vs Specificity.
- Table 3: the AK parameter was not included in the combined model, please clarify.
- At least in the figures captions please explicitly state which labels (0/1) correspond to pLGG and pHGG.
- Figure 5, patients at risk table must be included in the figure. Consider also a more detailed figure caption, particularly referring to the model and threshold used to classify low and high glioma grade.
- Lines 240-241: I suggest the following: … A random forest based model classifier was trained, hence applied…
- In the results paragraph correlations between survival and “DKI metrics” were reported, as well as the concordance index for Cox regression analysis considering MK as predictor (lines 252-257). As far as I understood from the method section I would have expected only Kaplan meyer (with log-rank test), based on pLGG/pHGG stratified with combined model. I suggest to clarify the method section to make the reported results consistent with methodology.
- At the end of the results paragraph the correlation between pLGG and pHGG (as predicted from the combined model) with OS and PFS were reported. It would be also interesting to report the actual correlation between true pLGG and pHGG with OS and PFS.
- The authors evaluated DKI and DTI metrics with respect to their respective mean values within the whole tumour volumes. It may be useful to add few considerations supporting this technical decision, instead of a more comprehensive histogram analysis within the DKI maps of the lesion volumes (e.g. previous studies investigating histogram analysis in glioma grading reported better performances for mean parameters, easiness of computation and evaluation enhanced their usability, better interpretability….).
- Line 370: include KFA in the acronyms list.
Author Response
Reviewer 1
Glioma are currently among the most common brain oncological lesions affecting paediatric patients. Tumour grading proved to have an overall strong correlation with clinical outcomes and predictive image-based models were extensively investigated in adult patients. The authors aimed at the development of an MR DKI-based grading model in paediatric glioma patients and the assessment of its correlation with survival metrics. The results are promising and consistent with literature data on adult patients, although further studies should be carried out on larger samples covering different tumour sites to confirm such results. Biases were pointed out in the discussion section.
I believe that this work is of high clinical interest, considering the potential impact on paediatric patients. I’m in favour of publication with minor revisions.
Thank you for your review, for the comments and for the important revisions you suggested. Below you can find the answer to the suggested revisions.
The manuscript is well written, with an overall good readability. Here follows few minor issues that need to be clarified or adjusted:
- Line 21: DKI acronyms should be added in the simple summary.
- Thank you, we added DKI acronyms in the simple summary as suggested,
- Considering the central importance of the statement I suggest to include a reference for the sentence line 62-63 (Introduction section).
- Thank you, we added a reference for the sentence line 62-63.
- Line 114: it is not clear if the 10 averages were acquired only for 0 b-value, or also for all the others (1000 and 2000) as they are expected to be more noisy.
- The 10 averages were acquired only for 0 b-values. This approach was chosen to allow us a scan time, which would be feasible for clinical practice. We also modified the text to include this information.
- Line 121: provide the Matlab version.
- Thank you, we added the Matlab version (9.2 R2017a).
- Please clarify the segmentation method and clinical decisions to account or not account for contrast-enhancing and non-contrast enhancing tumour regions. In particular, the whole-tumour volumes were segmented (line 126), ….. , contrast-enhancing and non-contrast enhancing tumour regions were segmented separately. How did you consider these two different segmentations? If you used them please clarify and provide results for them, otherwise I suggest a removal.
- Thank you for this question. In order to segment the whole tumor volume, we had to account for tumor heterogeneity in pediatric gliomas, which can contain non-enhancing components, enhancing components, or both. The classification tool of the Semiautomatic ITK SNAP segmentation yielded poor results when trying to segment together both tumor components. We therefore decided to segment separately the two tumor components (if both present) and included data from both components to account for intratumoral heterogeneity.
- The clinical approach to segmentation did not seem fully consistent between method section (lines 128-130) and discussion section (lines 304-317). Please clarify.
- Thank you for this observation, We edited the discussion to improve the consistency between the two sections.
- I suggest to include in the method section (lines 171-175), few characteristics of the predictive model (random forest) that was developed on the training set and applied in the testing set to derive the accuracy of such model. Also the input parameters should be mentioned (the one from the most significant model, the most significant metrics, all the DKI-derived metrics?).
Thank you for this question.
Thee random forest classifier was constructed through testing various hyperparameters, selecting the best ones in terms of predictive performance on the training set. The hyperparameters were manually setted as the following ones:
- “number of estimators” = 100 which describes the number of trees in the model
- “maximum depth” = 3 which describes the maximum depth of each tree in the forest.
A random seed was selected to guarantee the reproducibility of the results among different computers. Then it was selected a random seed which ensured a similar result for the model if used on different computers. We added this information in the text.
- In the method section (line 176-179) and also in the abstract (line 37), you stated that the combined model was tested for correlations with survival metrics. Did you consider the most accurate model (as reported in 160-161) or did you consider a priori the combined model for such analysis? I suggest to accordingly adjust this paragraphs.
Thank you for this observation. We confirm we tested for the most accurate model first and did not choose the combined model a priori. The results (AUC 0.946 for the combined model, (95%CI: 0.889-1)) proved this model the most accurate. We specified it in the abstract and revised the method section to improve consistency.
- Line 196: Table1, in bracket range or percentage.
Thank you, among the data presented in table 1 patient median age is expressed as a range, while all the other data within brackets are percentages. We added the percentage symbol where required to avoid confusion.
- Lines 199-202: considering ADC one of the DKI-metrics (although a DTI-metric) it is not consistent with previous sentence as higher in pLGG rather than in pHGG.
- Thank you, we corrected that by rephrasing the sentence.
- Figure 1, Figure 2, and Figure 4 should be consistent: True Positive rate vs False Positive Rate or Sensitivity vs Specificity.
- Thank you, we corrected figure 4 to make it more consistent. However, since reviewer 2 suggested to discard it, we added the revised fig. 4 in the supplementary material.
- Table 3: the AK parameter was not included in the combined model, please clarify.
- Thank you for this observation. The exclusion of the AK parameter from the model was not made by choice, but resulted as the output of the Elasticnet penalized regression model, which is fine-tuned to avoid overfitting by combining Ridge and LASSO penalties. Since AK and MK presented the highest AUC values we speculate there may be some information overlap between the two.
- At least in the figures captions please explicitly state which labels (0/1) correspond to pLGG and pHGG.
- Thank you, we edited the figures caption accordingly.
- Figure 5, patients at risk table must be included in the figure. Consider also a more detailed figure caption, particularly referring to the model and threshold used to classify low and high glioma grade.
- Thank you, we provide the patients at risk table together with DKI data in the Excel file we loaded as supplementary material. We also added information about the model in the caption of fig.5.
- Lines 240-241: I suggest the following: … A random forest based model classifier was trained, hence applied…
Thank you, we changed the phrase accordingly.
- In the results paragraph correlations between survival and “DKI metrics” were reported, as well as the concordance index for Cox regression analysis considering MK as predictor (lines 252-257). As far as I understood from the method section I would have expected only Kaplan meyer (with log-rank test), based on pLGG/pHGG stratified with combined model. I suggest to clarify the method section to make the reported results consistent with methodology.
- Thank you, we confirm the Kaplan Meyer analysis was based on pLGG/PHGG stratified with the combined model, we changed the results accordingly.
- At the end of the results paragraph the correlation between pLGG and pHGG (as predicted from the combined model) with OS and PFS were reported. It would be also interesting to report the actual correlation between true pLGG and pHGG with OS and PFS.
- Thank you, we reported the actual correlation between true pLGG and pHGG with OS and PFS. When performing the analysis, we realized we accidentally submitted the correlations with true pLGG and pHGG in the first version of the manuscript. We uploaded the correct KM curves on the revised version of the manuscript. We apologise for this mistake and thank the reviewer for giving us the opportunity to detect and correct it.
- The authors evaluated DKI and DTI metrics with respect to their respective mean values within the whole tumour volumes. It may be useful to add few considerations supporting this technical decision, instead of a more comprehensive histogram analysis within the DKI maps of the lesion volumes (e.g. previous studies investigating histogram analysis in glioma grading reported better performances for mean parameters, easiness of computation and evaluation enhanced their usability, better interpretability….).
- Thank you, actually utilising histogram analysis could have been more efficient in highlighting differences between pLGG and pHGG. We added this consideration in the discussion.
- Line 370: include KFA in the acronyms list.
Thank you, we included KFA in the acronyms list.

Reviewer 2 Report
The abstract has several abbreviations (ROC, SMOT) that are not unveiled and might be improved by focussing on the key points.
The Background information lacks the usual sensitivity / specifity data of currently available techniques: Biopsy, MRI +- contrast, DTI, other non-invasive techniques e.g. pet, spect etc.
How do you address the issue of progression free survival when you take the last follow-up. Doesn't that skew the analysis towards a lower number of PFS?
Would oncologists and surgeons base their decision on a sensitivity of 82.6% for a pHGG? Is this clinically meaningful or do you still need to do a biopsy? or would you base an indication for surgery on this?
Figure 4 is somewhat redundant to figure 3 and can be discarded.
It might be of interest to show a figure with an example of a 3D semi-automatic reconstruction of a tumour.
Have you tested this technique on any infratentorial tumours? This is where this kind of a technique could make a big difference (e.g. less surgical options, differentiating between gliomas and other common pediatric tumours)
Author Response
The abstract has several abbreviations (ROC, SMOT) that are not unveiled and might be improved by focussing on the key points.
Thank you for your observation. We added the Receiver Operating Characteristic (ROC) analysis and Synthetic Minority Oversampling Technique (SMOTE) abbreviation in the abbreviation list. We believe describing clearly these statistical analysis tools is important to our work. The characteristics and rationale of the SMOTE technique are described in detail in the Materials and Methods Statistical analysis paragraph.
The Background information lacks the usual sensitivity / specifity data of currently available techniques: Biopsy, MRI +- contrast, DTI, other non-invasive techniques e.g. pet, spect etc.
Thank you. Biopsy is widely regarded as the gold standard for diagnosis, we omitted this information in the Introduction but specified it in the materials and methods section. We added sensitivity and specificity data for DKI metrics in children and adults.
How do you address the issue of progression free survival when you take the last follow-up. Doesn't that skew the analysis towards a lower number of PFS?
Thank you, we decided for the last follow-up available at the reference date to collect as much clinical data on patient OS and PFS as possible. Patients, which did not present the event at the reference date or were lost during follow-up were censored, as usual. If this approach skew the analysis towards lower PFS values it would have been theoretically detrimental to our study (since we had a majority of pLGG in our cohort, it would have been more probable that a skew towards lower PFS would have occurred in this subgroup, minimizing the differences in PFS between pLGG and pHGG).
Patient at risk table, together with the results for DKI metrics for each patient, are provided in the Excel file submitted as supplementary material.
Would oncologists and surgeons base their decision on a sensitivity of 82.6% for a pHGG? Is this clinically meaningful or do you still need to do a biopsy? or would you base an indication for surgery on this?
Thank you for this question. First, the reported sensitivity threshold for the combined predictive model is 92.3%. The 82.6% percentage is referred to specificity (AUC: 0.946). We believe these results could be helpful in informing the following oncologist/ neurosurgeon decisions.
Second, we regard the implementation of DKI not as a substitute to biopsy, but rather as a tool for maximising the information that can be given to clinicians and neurosurgeons before therapeutic decisions.
Figure 4 is somewhat redundant to figure 3 and can be discarded.
Thank you for this observation, we discarded fig. 4 and renamed the remaining figures accordingly.
It might be of interest to show a figure with an example of a 3D semi-automatic reconstruction of a tumour.
Thank you for this observation. We created an additional figure which shows volume rendering of the tumor from the 3D semi-automatic segmentation (Fig.1).
Have you tested this technique on any infratentorial tumours? This is where this kind of a technique could make a big difference (e.g. less surgical options, differentiating between gliomas and other common pediatric tumours)
Thank you for your question. We included infratentorial hemispheric gliomas. We had no hemispheric cerebellar high-grade gliomas in our cohort, and we mentioned this as a limitation to the study in the discussion.
Our prediction model, however, was very accurate in predicting low-grade tumors with cerebellar hemispheric location.
